# EXPLORING POINTWISE SIMILARITY OF REPRESENTATIONS

## ABSTRACT

Representation similarity measures have emerged as a popular tool for examining learned representations. Many existing studies have focused on analyzing aggregate estimates of similarity at a *global* level, *i.e.* over a set of representations for N input examples. In this work, we shed light on the importance of investigating similarity of representations at a *local* level, *i.e.* representations of a single input example. We show that peering through the lens of similarity of individual data points can reveal previously overlooked phenomena in deep learning. Specifically, we investigate the similarity in learned representation of inputs by architecturally identical models that only differ in random initialization. We find that while standard models represent (most) inputs similarly only when they are drawn from training data distribution, adversarially trained models represent a wide variety of out-of-distribution inputs similarly, thus indicating that these models learn more "stable" representations. We design an instantiation of such a pointwise measure, named Pointwise Normalized Kernel Alignment (PNKA), that provides a way to quantify the similarity of an individual point across distinct representation spaces. Using PNKA, we additionally show how we can further understand the effects of data (*e.g.* corruptions) and model (*e.g.* fairness constraints) interventions on the model's representations.

## 1 INTRODUCTION

The success of deep neural network (DNN) models can be attributed to their ability to learn powerful representations of data that enable them to be effective across a diverse set of applications. However, the impressive performance of these models is often overshadowed by a variety of reliability concerns that arise when they are deployed in real-world scenarios (Geirhos et al., 2018; Hendrycks & Dietterich, 2019; Taori et al., 2020; Szegedy et al., 2013; Papernot et al., 2016; Athalye et al., 2018; Moosavi-Dezfooli et al., 2017; Angwin et al., 2016; O'neil, 2017). These concerns have led to a surge in interest in better understanding the internal representations of these models before deploying them (Alain & Bengio, 2016; Davari et al., 2022; Kriegeskorte et al., 2008). One promising line of research that offers a deeper understanding of model representations is representation similarity (Kornblith et al., 2019; Laakso & Cottrell, 2000; Raghu et al., 2017; Morcos et al., 2018). At their core, representation similarity measures provide an overall score that quantifies how a set of points are positioned relative to each other within the representation spaces of two models.

While aggregate measures have proved to be a useful tool to better understand many properties of deep learning (Nguyen et al., 2021a;b; Nanda et al., 2022; Raghu et al., 2021; Moschella et al., 2022), in this work we show that many other intriguing phenomena in deep learning can be understood by measuring the similarity of representations at the level of *individual data points*. Consider the well-studied case of two architecturally identical DNNs that only differ in random initialization. Prior works have independently concluded that two such models learn "similar" representations (indicated by a high aggregate representation similarity score on the test set) (Kornblith et al., 2019; Raghu et al., 2017). However, when analyzing similarity at the level of individual points, we find that *not all* points are represented similarly across these two models. Instead, we observe a few points whose representations obtain lower similarity scores. We refer to these as *unstable* points. We find that such unstable points hold some properties that can have implications for the models' performances on these points, *i.e.* models are more likely to disagree on the predictions for unstable points.

We further show how the use of a pointwise representation measure enables a deeper and better understanding of the connections between a model's representations and several other aspects, including its behavior and the impact of interventions on the acquired representations, both on the *data* employed (*e.g.* how changing the data distribution affects the representations of individual points) as well on the *model* itself (*e.g.* how training with fairness constraints changes representations of individual points).

To this end, we design an instantiation of such a pointwise representation similarity measure, which we call Pointwise Normalized Kernel Alignment (PNKA), that builds on the well-studied and broadly used Centered Kernel Alignment (CKA) (Kornblith et al., 2019) and assigns similarity scores to each point being evaluated across two distinct representations. Intuitively, for PNKA to assign a high similarity score to a point across two representation spaces, that point should be positioned similarly relative to the other points in both representations. Analogous to CKA, how to define the relative position of a point for PNKA can be changed flexibly by using the appropriate kernel function [1]. PNKA can be seen as a local decomposition of global representation similarity measures, by providing a distribution of similarity scores that when aggregated, provides an overall similarity estimation that is related to the aggregate measures broadly used today.

Our key contributions are summarized as follows:

- We highlight the importance of analyzing representation similarity at the granularity of individual data points. To this end, we design an instantiation of a measure, PNKA, that can provide similarities at a pointwise granularity.

- While the widely used aggregate representation similarity measures assign a high overall similarity score to the penultimate layer representations of two models that differ solely due to stochastic factors, *e.g.* in their random initialization, we show that not all individual inputs score equally highly. We call the points with lower representation similarity as *unstable*.

- Through a pointwise representation similarity measure (PNKA) we are able to investigate the properties that these points hold under different scenarios of data distribution shifts. We find that models are more likely to disagree on the predictions of unstable points. We also show that while non-robust models represent (most) points similarly only under an in-distribution context, adversarially trained models represent a wide variety of out-of-distribution samples similarly, thus indicating that these models learn more "stable" representations.

- Finally, using PNKA, we analyze how interventions to a model modify the representations of individual points. Applying this approach to the context of learning fair representations, we show that debiasing approaches for word embeddings do not modify the targeted group of words as expected, an insight overlooked by current evaluation metrics.

## 1.1 RELATED WORK

**Representation Similarity Measures.** Recently, approaches that compare the representational spaces of two models by measuring representation similarity have gained popularity (Laakso & Cottrell, 2000; Li et al., 2015; Wang et al., 2018; Raghu et al., 2017; Morcos et al., 2018; Kornblith et al., 2019). Raghu et al. (2017) introduced SVCCA, a metric based on canonical correlation analysis (CCA) (Hotelling, 1992), which measures similarity as the correlation of representations mapped into an aligned space. Morcos et al. (2018) build on this work by introducing PWCCA, another CCA-based measure that is less sensitive to noise. More recently, CKA (Kornblith et al., 2019) has gained popularity and has now been extensively used to study DNN representations (Nguyen et al., 2021a; Ramasesh et al., 2020; Raghu et al., 2019; 2021). CKA is based on the idea of first choosing a kernel and then measuring similarity as the alignment between these two kernel matrices. We take inspiration from this insight to propose PNKA. We refer readers to (Klabunde et al., 2023) for a comprehensive overview of similarity measures.

**Understanding Representations of Individual Data Points using Neighbourhoods** The broad idea of comparing nearest neighbors of instances in the representation space has been introduced in prior works, albeit for different motives, *e.g.* changes in linguistic styles (Hamilton et al., 2016), analyzing node embeddings (Wang et al., 2020), and for robust prediction (Papernot & McDaniel, 2018). While

---

[1]Similar to the CKA paper we use a linear kernel for all our experiments.

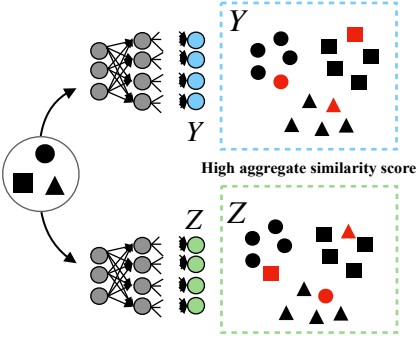
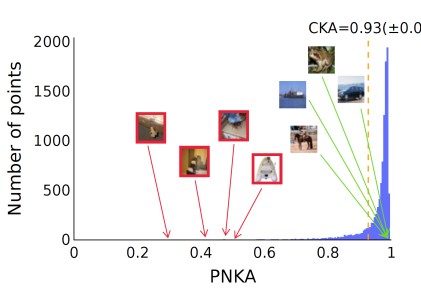

(a) Examples of global × local measures.

(b) Distribution of similarities.

Figure 1: *Left*: An illustrative example showing that aggregate similarity measures over representations $Y$ and $Z$ are not able to provide insights into the distribution of similarity scores at the level of individual points. In this example, a pointwise measure shows that while the majority of points (in black) are positioned highly similarly relative to the other points, a minority of points (in red) are scattered highly dissimilarly. *Right*: Distribution of similarity scores for CIFAR-10 test set. Results are an average over 3 runs, each one containing two models trained on CIFAR-10 with the same architecture (ResNet-18) but different random initialization. While most of the points are similarly represented (which agrees with CKA score), some are less similarly represented.

our method is inspired by the higher-level idea of comparing neighborhoods across representations, we differ significantly from these works since we offer a concrete measure of *pointwise* similarity that is general-purpose and can be broadly applied to understand many phenomena in deep learning, across different data modalities. Recent work by Shah et al. (2023) proposes a method to estimate the contribution of individual points to learning algorithms. However, their work is mainly focused on understanding what features of inputs are encoded in the representations and does not evaluate the similarity of representations. Instead, in our work, we focus on showing the importance of analyzing whether two models represent individual inputs similarly. Work by Moschella et al. (2022) also relates to ours as their proposed model stitching method resembles our proposed measure (PNKA). However, we note here that the goal, contributions, and assumptions made in their paper differ drastically from ours. More importantly, their method assumes that the angles between elements of the latent space are kept the same for *all* elements, which we show in Section 4 as not being the case.

## 2 WHY STUDY REPRESENTATION SIMILARITY AT FINER GRANULARITY

Previous studies have primarily focused on inspecting the aggregate-level representation similarity of DNNs. As a consequence, these studies do not provide insights into the distribution of similarity scores at the granularity of individual data points. We illustrate this in Figure 1a, where we compare representation spaces of two models, namely $Y$ and $Z$. The majority of points (in black) are positioned highly similarly relative to the other points, in both representations, while a minority of points (in red) are positioned highly dissimilarly. We need fine-grained pointwise similarity scores to enable us to distinguish between these *stably* represented (black) and *unstably* represented (red) points.

In Figure 1b, we demonstrate the need for such a fine-grained measure with a concrete example. Figure 1b shows the distribution of pointwise similarity scores, on the CIFAR-10 test set, for two ResNet-18 models that only differ on their random initialization, but are otherwise trained using the same procedure on CIFAR-10 [2]. We also illustrate some data points sampled at different points in the distribution. We see that most of the points exhibit high similarity scores, which also aligns with the high CKA score obtained [3]. However, there exist some (*unstable*) points in the tail of the distribution with lower representation similarity scores. Identifying unstable points whose representational stability is impacted solely by stochastic factors (*i.e.* randomness) within the training process is not

---

[2]More information on training details as well as test set accuracies can be found in Appendix A.

[3]We expand this analysis to other architectures and datasets in Appendix B.

only valuable but also crucial. As we show later in Section 4, these points are not only more likely to originate from out-of-distribution sources but are also prone to higher misclassification rates.

Finally, we note that some prior studies using aggregate similarity measures implicitly assume that representational instability arising from randomness in the training procedure would be limited to very few points. The original CKA paper (Kornblith et al., 2019) claimed that two models that differ only in their random initialization would learn highly similar representations at the penultimate layer, without qualifying that the observation holds true only for inputs drawn from the training data distribution. This observation has since been even proposed as a sanity check to audit different representation measures, *e.g.* for Ding et al. (2021) a reliable similarity measure *must* assign high similarity to representations from models that only differ in random initialization. As we show in Section 4, the stability of learned representations for models with different random initializations is strongly influenced by other factors such as whether models use robust or standard learning procedures. Thus, the validity of conducting such a sanity check becomes questionable.

## 3 MEASURING POINTWISE REPRESENTATION SIMILARITY

In order to analyze representation similarity at a *local* level, we design an instantiation of a pointwise representation similarity measure, named Pointwise Normalized Kernel Alignment (PNKA), which builds on the well-studied and broadly used Centered Kernel Alignment (CKA).

**Notation.** We denote by $Y \in \mathbb{R}^{N \times d_1}, Z \in \mathbb{R}^{N \times d_2}$ two sets of $d_1$ and $d_2$ dimensional representations for a set of $N$ inputs, respectively. We assume that $Y$ and $Z$ are centered column-wise, *i.e.* along each dimension. We aim to measure how similarly the $i$-th point is represented in $Y$ and $Z$. We denote a pointwise similarity measure between representations $Y$ and $Z$ for point $i$ by $s(Y, Z, i)$.

**Formally defining PNKA.** To design PNKA, we leverage the simple, but powerful insight from prior works, which states that while we cannot directly compare similarity *across* representations, we can do so *within* the same representation (Kornblith et al., 2019; Kriegeskorte et al., 2008). Therefore, to determine whether the representations $Y_i$ and $Z_i$ of point $i$ are similar, we can first compare how similarly $i$ is positioned relative to all the other points within each representation. We then compare the relative position of $i$ across both representations.

More formally, given a set of representations $Y$ and a kernel $k$, we can define a pairwise similarity matrix between all $N$ points in $Y$ as $K(Y)$ with $K(Y)_{i,j} = k(Y_i, Y_j)$. In our work, we use linear kernels, *i.e.* $k(Y_i, Y_j) = Y_i \cdot Y_j^\top$, but other kernels, *e.g.* RBF (Kornblith et al., 2019) could be used as well. We leave the exploration of other types of kernels for future work. Given two similarity matrices $K(Y)$ and $K(Z)$, we measure how similarly point $i$ is represented in $Y$ and $Z$ by comparing its position relative to the other points. To this end, we define

$$\text{PNKA}(Y, Z, i) = \cos(K(Y)_i, K(Z)_i) = \frac{K(Y)_i^\top K(Z)_i}{||K(Y)_i|| \, ||K(Z)_i||}, \quad (1)$$

where $K(Y)_i$ and $K(Z)_i$ denote how similar point $i$ is to all other points in $Y$ and $Z$, respectively. We use cosine similarity to compare the relative positions across representations for two reasons. First, cosine similarity provides us with normalized similarity scores for each point. Second, by normalizing by the length of the similarity vectors $K(Y)_i$ and $K(Z)_j$, we compare the *relative* instead of the absolute similarity of points, *i.e.* how similar point $i$ is represented relative to points $j$ and $j'$. PNKA can also be extended into an aggregate version, that has empirically shown to be correlate with CKA (Kornblith et al., 2019) (Appendix C.1), by computing

$$\overline{\text{PNKA}}(Y, Z) = \frac{1}{N} \sum_{i=1}^{N} \text{PNKA}(Y, Z, i), \quad (2)$$

**Computing PNKA with stable reference points.** As PNKA works by comparing how a point is positioned relative to other reference points across two representation spaces, one may wonder if the reference points themselves should be required to have stable representations. For instance, in Figure 1a, computing PNKA scores using unstable (red) points as reference points might yield low similarity scores for all points. To this end, one can construct a particular case of PNKA, restricting the set of $N$ reference points to $L$ *stable* points. We establish that reference points in this context must adhere to two essential properties: (1) *stability*: points should remain stably positioned relative

to each other, *i.e.* have high $\overline{\text{PNKA}}$ amongst themselves, and (2) *spatial diversity*: points should be well-distributed in the representation space, *i.e.* points should not be collapsed. We show in Appendix C.2 that these two properties hold for our choice of reference points. The reference points can come from the training set or as a subset of the test set distribution ($L \subseteq N$, where $L = N$ is the general case previously presented). In the experiments of the following section, we draw $L = 1,000$[4] reference points from the training set, *i.e.* we compute the relative position of the $N$ test set points with respect to a subset of $L$ stable and spatially diverse points from the training set.

Formally, given the representations of points $A \in \mathbb{R}^{N \times d_1}, C \in \mathbb{R}^{N \times d_2}$, and respective reference points $B \in \mathbb{R}^{L \times d_1}, D \in \mathbb{R}^{L \times d_2}$, from two models with dimensions $d_1$ and $d_2$, respectively, we define a pairwise similarity matrix as $K(A, B)$ with $K(A_i, B_j) = k(A_i, B_j)$. Thus, in this specific case PNKA is defined as

$$\text{PNKA}(Y, Z, i) = \cos(K(A, B)_i, K(C, D)_i) = \frac{K(A, B)_i^\top K(C, D)_i}{||K(A, B)_i|| \, ||K(C, D)_i||}, \tag{3}$$

where $K(A, B)_i$ and $K(C, D)_i$ denote how similar point $i$ is to the $L$ reference points in each of the models.

**Properties.** We empirically show that PNKA holds important properties (Kornblith et al., 2019), such as invariance to both orthogonal transformations and isotropic scaling (Appendix C.3). We also empirically show that PNKA captures the overlap of neighbors across two representations and that if the PNKA score of point $i$ is higher than that of $j$, then there is a higher chance that $i$'s nearest neighbors overlap more across representations $Y$ and $Z$ than those of $j$ (Appendix C.4).

# 4 USING POINTWISE ANALYSIS TO UNDERSTAND DATA INTERVENTIONS

In this section, we use PNKA to investigate the properties of *unstable* points, *i.e.* points represented less similarly, between models that differ solely due to their random initialization, and analyze if these points possess some distinct properties. We deliberately chose to focus on comparing representations of models that differ on random initialization as in this scenario, unstable points represent inputs whose representations are heavily influenced by random chance, and using such unstable representations for downstream tasks can be worrisome. We analyze the (in)stability of representations under three scenarios: (1) *in-distribution data points* (Section 4.1), *e.g.* the test set, which exemplifies a usual scenario where the model will be used for the same downstream task that it has been previously trained for; (2) *subset of data points is out-of-distribution* (Section 4.2) which might illustrate a practical scenario in which individuals seek to evaluate models on "in-the-wild" data while already possessing a set of trusted (in-distribution) data points; (3) *all data points are OOD* (Section 4.3), which portrays a scenario where the features of the models might be used for a different task than the model was previously trained for, *e.g.*, transfer learning. In the remainder of this section, we report an average PNKA score over 3 runs of two models trained on CIFAR-10 (ResNet-18 (He et al., 2016)) differing only in their random weight initialization.

## 4.1 MODELS MORE LIKELY TO DISAGREE ON UNSTABLE POINTS

We first examine unstable points for inputs that fall within the training distribution, *i.e.* CIFAR-10 test set. We also expand this analysis to CIFAR-10.1 (Recht et al., 2018), which attempts to construct another CIFAR-10 test set, closely following the methodology of the original dataset, but which has been shown to cause a significant drop in accuracy ($4 - 10\%$). Given that unstable points exhibit greater dissimilarity across models trained with different initializations, a reasonable hypothesis is that these models will be more prone to disagreeing on the predictions for these unstable points. In Figure 2 we show the percentage of instance predictions on which the models agree, relative to their ranked similarity score. The points were first sorted according to their similarity scores, with the leftmost end (0) representing the group with the lowest scores and the rightmost end (9) representing the group with the highest scores, and then grouped into deciles, with each bar representing 10% of the total points in the test set. The vertical dotted line shows the aggregate scores ($\overline{\text{PNKA}}$) for each group. We can see that the fraction of points whose predictions the models disagree on are mainly at the tail of the distribution, *i.e.* being less similarly represented, for both CIFAR-10 and CIFAR-10.1 test sets. In Appendix D.1 we show the same pattern for other choices of architecture and dataset. We

---

[4]10% of the total amount of test set points of CIFAR-10 and CIFAR-100 datasets.

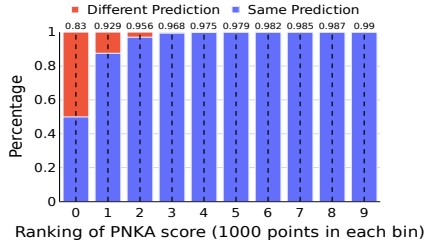

(a) Ratio of agreement (CIFAR-10).

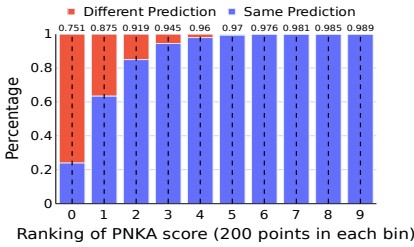

(b) Ratio of agreement (CIFAR-10.1).

Figure 2: Percentage of instance predictions on which the models agree, relative to their ranked similarity score, for both CIFAR-10 (a) and CIFAR10.1 (b) test sets. The $x$-axis represents groups of points sorted based on their pointwise representation similarity according to PNKA, with each group (bar) containing 10% of the total amount of instances. The $y$-axis represents the fraction of those points on which models agree (blue) or disagree (red). The vertical dotted line shows the aggregate scores ($\overline{\text{PNKA}}$) for that group. Results are averaged over 3 runs, each one containing two models trained on CIFAR-10 with different random initialization. The more unstable a point is, *i.e.* lower its representation similarity, the more likely models are to disagree on its prediction.

also show in Appendix D.1.2 that these points are not only classified in different ways but that most of them are misclassified as well. Lower accuracy is to be expected since if two models disagree on a prediction, at most one of them can be correct. This finding also aligns with previous work on calibration Baek et al. (2022); Jiang et al. (2021); Garg et al. (2022) which uses a model's outputs to detect which instances are more likely to be misclassified. Therefore, unstable points are those for which models exhibit the greatest prediction disagreement and incorrect predictions.

### 4.2 OUT-OF-DISTRIBUTION POINTS MORE LIKELY TO HAVE UNSTABLE REPRESENTATIONS

Next, we examine the case where some points do not come from the training distribution. To inspect that, we perturbed $p\%$ of the test set points with naturally occurring perturbations, *e.g.* blurring, color jitter, and elastic transformation. We then compute PNKA on the test set with $p\%$ perturbed and $1 - p\%$ non-perturbed (*i.e.* originally from the test set) points for models that differ in their random initialization. We hypothesize that the representations of models are similar for points that have a high likelihood under the models' training distributions, but that the representations of models will be dissimilar on OOD points. In Figure 3 we show the percentage of perturbed instances, relative to their ranked similarity scores. As previously, points were sorted according to their similarity score and then grouped into deciles. We use $p = 10\%$ and show that, for different types of perturbations, perturbed points are more likely to obtain lower similarity scores compared to non-perturbed (*i.e.* in-distribution) points. Thus, under this scenario, unstable points are more likely to be OOD than points with higher representation similarity. We expand this analysis for other choices of $p$, architectures, and datasets in Appendix D.2.

### 4.3 ROBUST MODELS ARE LESS INFLUENCED BY STOCHASTIC FACTORS

Finally, we investigate the extreme scenario of data distribution shift, where all the samples are out-of-distribution, *i.e.* $p = 100\%$. Prior work (Ding et al., 2021; Davari et al., 2023; Nguyen et al., 2021a;b; McCoy et al., 2019) has employed global measures of representation similarity to examine models' representations when exposed to out-of-distribution (OOD) data. It has been observed that these models exhibit dissimilar representations, even when the sole difference lies in their random initialization. Under this scenario, we also study unstable points for adversarially trained (*i.e.* robust) models as they are trained to be more resilient to adversarial examples, *i.e.* samples that are slightly perturbed to alter the model's behavior. For both types of models, we again compute the pointwise representation similarity between models that differ only in random initialization. Figure 4 shows the similarity scores distribution for both robust and non-robust models trained on CIFAR-10 and evaluated under their original distribution, CIFAR-10 test set (Figure 4a), as well as two different distribution shifts: CIFAR-100 (Figure 4b) and images with complete random noise (Figure 4c). We

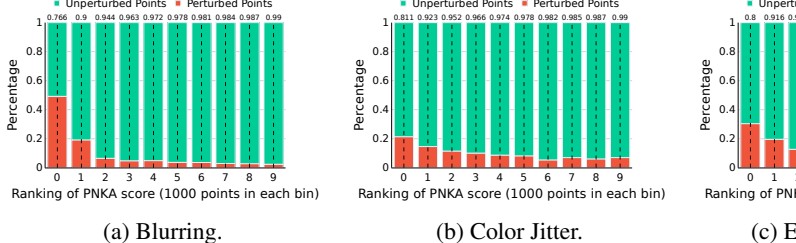

(a) Blurring.     (b) Color Jitter.     (c) Elastic Transform.

Figure 3: Percentage of perturbed instances, relative to their ranked similarity score. The $x$-axis represents groups of points sorted based on their pointwise representation similarity according to PNKA, with each group containing 10% of the total amount of instances. The $y$-axis represents the fraction of the points that are perturbed (red) or not perturbed (green). The vertical dotted line shows the aggregate scores ($\overline{\text{PNKA}}$) for that group. We consider three possible perturbations: (a) blurring, (b) color jitter, and (c) elastic transformation. Results are over CIFAR-10 test set instances, averaged over 3 runs, each one containing two models trained on CIFAR-10 with different random initializations. Note that the more unstable a point is, *i.e.* lower the representation similarity, the more likely a point is to be OOD.

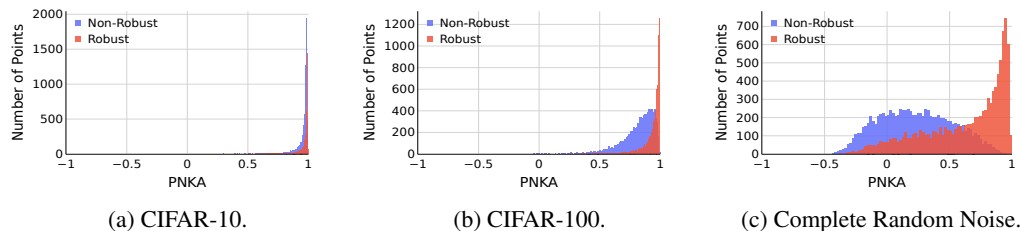

(a) CIFAR-10.     (b) CIFAR-100.     (c) Complete Random Noise.

Figure 4: Distribution of similarity scores for standard (non-robust) models (blue) and adversarially trained (robust) models (red). Results are averaged over 3 runs, each one containing two models trained on CIFAR-10 with different random initialization. The pointwise similarity scores are shown for (a) CIFAR-10 test set (in-distribution), as well as (b) CIFAR-100 and (c) complete random noise. While standard models represent (most) inputs similarly only when they are drawn from training data distribution (left-most figure), adversarially trained models represent a wide variety of out-of-distribution inputs similarly, thus indicating that these models learn more "stable" representations.

can see that under a similar training distribution (Figure 4b), both robust and non-robust models have similar PNKA distributions. However, as we use points further away from the distribution, the robust models seem to obtain more stable representations than the non-robust model. Even for complete random noise, the robust model represents several points similarly, *i.e.*, PNKA score $> 0.9$. This suggests that robust models learn more "stable" representations across a wide variety OOD data. We expand this analysis to other types of OOD data, as well as models trained on other datasets in Appendix D.3.

## 5   USING POINTWISE ANALYSIS TO UNDERSTAND MODEL INTERVENTIONS

Pointwise representation similarity can also be a useful tool to better understand the effects of interventions on a model. We can use PNKA to compute pointwise similarity scores between the representations of the original and the modified (*i.e.* intervened) models and analyze the inputs that are most affected by the intervention. We showcase the use of PNKA in the context of interventions to learn fair ML models.

An important goal of the fair ML literature is *non-discrimination*, where we attempt to mitigate biases that affect protected groups in negative ways (Angwin et al., 2016; O'neil, 2017). A popular approach to achieve non-discrimination is through learning debiased or fair representations (Zemel et al., 2013; Creager et al., 2019; Louizos et al., 2015). These approaches transform or train model

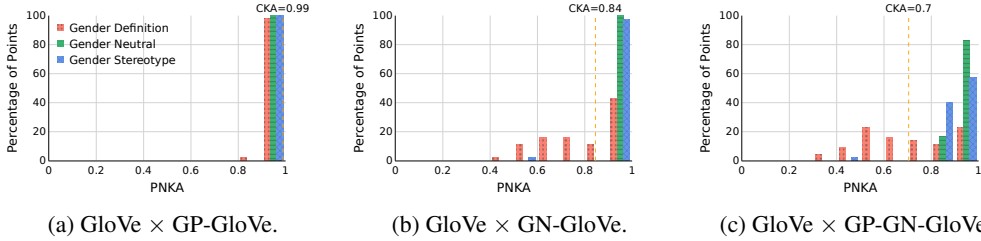

(a) GloVe × GP-GloVe.    (b) GloVe × GN-GloVe.    (c) GloVe × GP-GN-GloVe.

Figure 5: Distribution of PNKA scores per group of words for SemBias dataset (Zhao et al., 2018). We compare the baseline (GloVe) model and its debiased versions. Words with the lowest similarity scores are the ones that change the most from the baseline to its debiased version. Surprisingly, across all debiased embeddings, the words whose embeddings change the most are the gender-definition words.

representations in a way that minimizes the information they contain about the group membership of inputs. However, today, we often overlook how the interventions targeting (macro-)group-level fairness affect representations at the (micro-)individual-level and whether the changes in individual point representations are desirable or as intended. By applying PNKA to the original and the debiased representations, we can understand the effects of the debiasing intervention at the level of individual inputs, and analyze the inputs whose representations underwent the biggest change. We demonstrate how this ability can be leveraged in the context of natural language word embeddings to investigate whether the debiasing approaches indeed work as intended.

**Approaches to debias word embeddings:** Many word embedding approaches have been found to produce biased representations with stereotypical associationss (Bolukbasi et al., 2016; Gonen & Goldberg, 2019; Zhao et al., 2018), and several methods have been proposed with the goal of reducing these stereotypical biases (Bolukbasi et al., 2016; Gonen & Goldberg, 2019; Zhao et al., 2018; Kaneko & Bollegala, 2019). In this work, we choose two approaches with the goal of using PNKA to analyze whether debiasing successfully decreases stereotypical associations. Both debiasing techniques are based on the original GloVe (Kaneko & Bollegala, 2019): (1) Gender Neutral (GN-)GloVe (Zhao et al., 2018) focuses on disentangling and isolating all the gender information into certain specific dimension(s) of the word vectors; (2) Gender Preserving (GP-)GloVe (Kaneko & Bollegala, 2019) targets preserving non-discriminative gender-related information while removing stereotypical discriminative gender biases from pre-trained word embeddings. The latter method can also be used to finetune GN-GloVe embeddings, generating another model namely, GP-GN-GloVe.

**Evaluation of debiased word embeddings:** In order to evaluate the impact of the debiasing methods, both GP- and GN-GloVe use the SemBias dataset (Zhao et al., 2018). Each instance in SemBias consists of four word pairs: a *gender-definition* word pair (*e.g.* "waiter - waitress"), a *gender-stereotype* word pair (*e.g.* "doctor - nurse"), and two other word-pairs that have similar meanings but no gender relation, named *gender-neutral* (*e.g.* "dog - cat"). The goal is to evaluate whether the debiasing methods have successfully removed stereotypical gender information from the word embeddings, while simultaneously preserving non-stereotypical gender information. To this end, GP- and GN-GloVe evaluated how well the embeddings can be used to predict stereotypical word pairs in each instance of the SemBias dataset. The details and results of this prediction task are in Appendix E.1. The evaluation shows that GP-Golve embeddings offer only a marginal improvement, while GN- and GP-GN-GloVe embeddings offer substantial improvement at the prediction task.

**Using PNKA to understand debiased word embeddings:** We applied PNKA to the original and the debiased GloVe embeddings to examine whether the methods are indeed reducing bias as claimed. Figure 5 shows the distribution of PNKA scores for words in SemBias dataset grouped by their category (*i.e.*, gender defining, gender neutral, and gender stereotype). We highlight two observations. First, GP-Glove representations are very similar to GloVe (Figure 5a) for almost all of the words, whereas GN-Glove (Figure 5b) and GP-GN-GloVe (Figure 5c) considerably change the representations for a subset set of the words. This observation aligns well with results of prior evaluation which found that GP-GloVe achieves similar results to GloVe, while GN-Glove and GP-GN-Glove achieve better debiasing results. Second, Figure 5 also shows that across all three debiasing methods, the

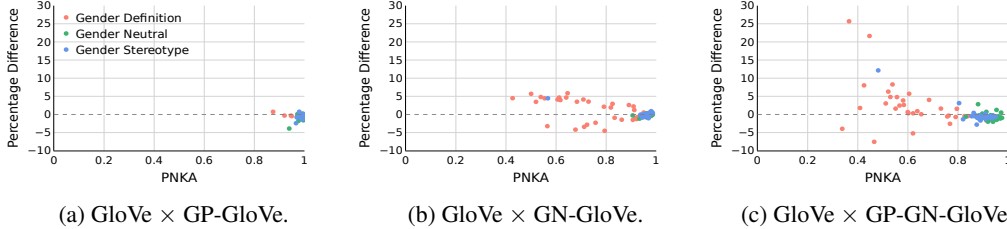

(a) GloVe × GP-GloVe.  (b) GloVe × GN-GloVe.  (c) GloVe × GP-GN-GloVe.

Figure 6: Relationship between PNKA scores and percentage change in magnitude of the projection onto the gender vector from the baseline GloVe. A positive change indicates an increase in magnitude along the canonical gender direction. Word embeddings that change their gender information are the ones that obtain low PNKA scores.

words whose embeddings change the most are the gender-definition words. Note that this observation is in complete contradiction to the expectation that with debiasing, the embeddings that would change the most are the gender-stereotypical ones, while the embeddings that would be preserved and not change are the gender-definitional ones. Put differently, the pointwise similarity scores suggest a very different explanation for why GN-GloVe and GP-GN-GloVe achieve better debiasing evaluation results over SemBias dataset: rather than remove gender information from gender-stereotypical word pairs, they are amplifying the gender information in gender-definition word pairs, resulting in better performance in distinguishing gender-stereotypical and gender-definition word pairs.

We confirm our alternate explanation by measuring for each word how much its embedding changed in terms of gender information, when compared to the original GloVe embedding, by projecting it onto the canonical gender vector $\overrightarrow{he}$ - $\overrightarrow{she}$ (more in Appendix E.2), generating the percentage difference in magnitude. Figure 6 shows that the GN-GloVe and GP-GN-GloVe debiasing methods primarily amplify the gender information in gender-definition words, rather than reduce it for gender-stereotype words. In fact, the words that change their gender information the most are the low-similarity ones. This analysis illustrates how pointwise similarity scores can offer new insights, trigger new investigations, and lead to a better understanding of the effects of model training interventions.

## 6 DISCUSSION

In this work, we demonstrate the power of investigating representations at the level of individual data points. First, we show that not all data points obtain a high similarity score, even for models that differ solely due to differences in random weight initialization. Under this context, we define the lower similarity points as unstable. We then investigate some of the characteristics of unstable points, including a higher likelihood of model prediction disagreements and the possibility that these points might be out-of-distribution. We then show that while standard (*i.e.* non-robust) models represent (most) inputs similarly only when they are drawn from the training data distribution, adversarially trained (*i.e.* robust) models exhibit higher representation similarity for a broader range of out-of-distribution points. This finding suggests that robust models learn more "stable" representations. Finally, we use the context of fairness to show that pointwise similarity measures can be a useful tool for understanding which individuals are most affected by model interventions, thus shedding light on the internal characteristics of such modifications. A limitation of our work lies in the restricted consideration of only a few model variations.

**Other applications of pointwise representation similarity analysis.** Employing pointwise representation similarity measures unveils several intriguing directions for exploration. For instance, one could examine differences in different architectures through the lens of similarity of individual points. An initial exploration in this direction is presented in Appendix F. Another promising line of work could potentially analyze how points are represented (dis)similarly across different layers of a neural network. We offer an initial analysis in this direction in Appendix G. Finally, one can use PNKA to delve deeper into the understanding of individual neurons within a neural network layer. We provide an initial analysis of the influence of single neuron units on pointwise representation similarity in Appendix H.

**Reproducibility.** We run all our experiments using publicly available, open-source frameworks, architectures and datasets. Thus, all our results can be seamlessly reproduced. We also attach our code to aid reproducibility. To ensure correctness, we also report all our results over 3 random seeds. All other details about pre-processing, learning rate, epochs, model architectures, and more information can be found in Appendix A and are also included in our attached code.

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
