# OpenReview forum: "Exploring Pointwise Similarity of Representations"
_ICLR.cc/2024/Conference — Submitted to ICLR 2024_

### Official Review · Reviewer_vKT7 · 2023-10-30

**Soundness:** 2 fair
**Presentation:** 2 fair
**Contribution:** 2 fair
**Rating:** 5
**Confidence:** 3

**Summary:**

The paper proposes new similarity measure between two representation spaces. The main idea of the new measure lies in assessing similarity locally, i.e., by comparing neighbourhoods of points between representation spaces.

**Strengths:**

The development of reliable similarity measures between representations is important and ongoing direction in the modern deep learning. Authors suggest studying similarity from the perspective of neighbourhoods of points and examine the proposed measure in various experiments.

**Weaknesses:**

My main concern is that authors propose a new metric which is build on CKA and CKA is known to have pitfalls (i.e. [1]). Thus, the very important part of suggesting a new metric is studying the pitfalls of the new measure and understanding the differences from the existing metrics. In the current version of the paper I did not see such investigation.

Also, for the similarity measure, it is important to understand the context in which we are using them. Authors argue that studying neighbourhoods might make sense, but does not discuss in which context it is important. For example, authors show that PNKA as CKA also shares such properties as the invariance to orthogonal transformations and to isotropic scaling, but again sometimes it can be beneficial, sometimes not, depends on the context.

Thus, deeper understanding of pitfalls and studying application areas are important to prevent careless use of the new metric by the community.

[1] Davari et al. Reliability of CKA as a Similarity Measure in Deep Learning. ICLR 2023

**Questions:**

In general, I would like to see the additional analysis as mentioned in the Weaknesses part.

---

> ### Author Response · Authors · 2023-11-17
>
> We appreciate the feedback provided by the reviewer and the insightful comments.
>
> We first would like to emphasize that the major contribution of our work, as mentioned in the introduction of the paper, is to highlight the importance of analyzing representation similarity at the granularity of individual data points. To this end, we design an instantiation of such a measure, PNKA, however, other instantiations are also possible, and would probably be useful under different contexts. As the reviewer pointed out, the interpretation of invariances or sensitivities in (similarity) measures, like CKA and PNKA, is context-dependent.
>
> Nevertheless, we recognize the importance of addressing the concerns raised regarding PNKA and its relationship with CKA, especially in light of the pitfalls associated with CKA, as discussed in [1].
>
> ## Important Invariances in Pointwise Representation Similarity Measures
> In the case of pointwise representation similarity measures, the main goal is to assess the similarity of *individual* data points across two representations. Therefore, the pointwise measure should be sensitive to changes in the relative position of points.
>
> The invariances described in the CKA paper, such as invariance to orthogonal transformations and isotropic scaling, are important in the context of pointwise representation measures. Rotating or uniformly scaling representations does not alter the relative position of the points, i.e. the result should be the same (in a geometric perspective) as the original one. Thus, we show that PNKA obtains invariance to such transformations.
>
> ## Important Sensitivities in Pointwise Representation Similarity Measures
> As shown in [1], CKA is sensitive to outliers and transformations preserving linear separability. While CKA's sensitivity to these factors may be perceived as a limitation in some contexts, it becomes an important property in *pointwise* representation similarity analysis.
>
> Consider the sensitivity to outliers, as outlined in [1], where two representations that are identical in all aspects except for the fact that one of them contains an outlier, i.e. a representation further away from the others. The fact that CKA's value decreases with increasing differences in outlier position indicates higher dissimilarity between representations.  However, in the context of pointwise representation similarity analysis, sensitivity to outliers is not a weakness but a valuable property. Outliers, defined as points that change their positions relative to others [1], should indeed be detected by the pointwise measure. As we illustrate in Section 4.2 of our paper, out-of-distribution  (OOD) points, i.e. outliers, are more likely to obtain unstable representations, i.e. exhibit lower representation similarity. Therefore, PNKA's responsiveness to outliers can be employed to infer such properties from data points.
>
> We hope we have suitably addressed all of the reviewer’s concerns and we would gladly go into more detail if there are any remaining questions.
>
> References:
>
> [1] Davari et al. Reliability of CKA as a Similarity Measure in Deep Learning. ICLR 2023

---

### Official Review · Reviewer_UEwp · 2023-10-30

**Soundness:** 3 good
**Presentation:** 3 good
**Contribution:** 2 fair
**Rating:** 5
**Confidence:** 4

**Summary:**

the submission proposed a point-wise normalized kernel alignment for measuring the similarity of a pair of vector representations that are produced by two trained neural networks on a single data point. The core concept which the proposed similarity score draws inspiration from is the assumption that similar vector representations should have similar neighbours, so we can directly measure the similarity of neighbours of the same point in two representation spaces.

Through experiments, they showed that trained models are likely to disagree on points with representations that are not so similar, and robust models are likely to agree more since they produce similar representations.

**Strengths:**

1. the proposed similarity score is well-motivated, and easy to implement.

2. the experiments show evidence of the effectiveness of the proposed score.

**Weaknesses:**

I have several questions regarding the usefulness of the proposed scores.

1. if the assumption is that the neighbours of a single point in two representation spaces matter in the construction of useful similarity scores, then I think an easy and effective approach would be Jaccard distance, and its variants that take distances into consideration.  I wonder how the proposed approach compares to Jaccard distance.

2. it seems natural that models tend to disagree on misclassified data points, so if that is the case, we would then only need to look at the misclassified points as the unstable points rather than using the proposed similarity score to determine the unstable ones?

3. since the comparison is now conditioned on a single data point along with its reference points, when two models are presented to us, how do we determine which model to use? The submission mentioned transfer learning as a use case, but it seems relatively non-trivial to me in terms of how we use the score in selecting the better pre-trained model to transfer from.

**Questions:**

n/a

---

> ### Author Response · Authors · 2023-11-17
>
> We thank the reviewer for the constructive feedback and comments. Our review responses are placed below.
>
> ## Jaccard distance vs PNKA
> Our primary goal in this work is to highlight the importance of analyzing representation similarity at the granularity of individual data points. To achieve this goal, we designed *one* instantiation of a pointwise measure, PNKA, that can provide similarities for individual points across representations. However, we note here that other pointwise measures can also be employed.
>
> In **Appendix C.4**, we empirically show that PNKA is correlated with the overlap of $k$ nearest neighbors, i.e., if the PNKA score of point $i$ is higher than that of another point $j$, then there is a *higher chance* that $i$’s nearest neighbors overlap more across the representation space of two models than those of $j$. We observed, however, that the correlation depends on the choice of $k$, and varies across architectures/datasets. Similar results, showing the relationship between PNKA and the Jaccard similarity coefficient, are presented in **Appendix C.5**, and are also shown to depend on the choice of $k$.
>
> Given the observed variability in the results due to the choice of $k$, we ran an alternative analysis of unstable points using the Jaccard coefficient and appended the results in **Appendix D.1.1** of the updated version. We ran the experiment for four different $k$ values (k=250, 500, 1000 and 2000), different architectures (ResNet-18, VGG-16), and datasets (CIFAR-10, CIFAR-100).
>
> From the results, we can infer that *the relationship between unstable points and the misclassification rate is highly affected by the choice of $k$ in the Jaccard coefficient*, and the *decision of which $k$ to choose from is not trivial*. Moreover, the optimal $k$ for one architecture and dataset does not generalize to other architectures and datasets. We also show in **Appendix C.2.1** that the choice of reference points, and number of landmarks, does not heavily influence the results for PNKA. Thus, Jaccard coefficient is not able to provide the same insights as PNKA.
>
> ## Misclassified instances as unstable points
> While it's true that models tend to exhibit differences in their predictions for dissimilarly represented points, it's essential to note that the analysis of misclassification is a task-dependent analysis that requires the annotation of ground-truth labels. PNKA, on the other hand, as a representation similarity measure, offers a valuable approach to understanding model behavior without relying on ground-truth labels, by analyzing the internal representations themselves. Thus, by applying PNKA, one can already identify unstable points without a task in mind, and without access to any annotation. The stability of a point, as defined in the paper, is an inherent property of both the input data and the models under analysis, and is *not* task-dependent. For example, determining whether a point is misclassified relies on the exact classification task, e.g. 100-class classification or a coarser 20-super-class classification for CIFAR-100 dataset. In each case, the set of misclassified points may vary, mainly due to the higher level of granularity in the latter task. Thus, using misclassification rates to define unstable points could result in shifting sets of unstable points, even when the data points and models remain unchanged. We, however, argue that a point's instability remains consistent, irrespective of the specific application.
>
> ## Using PNKA scores
> In Section 5, we show that PNKA can be applied for model selection by illustrating how a model designer can compare different model representations to check which one may be more desirable for a specific downstream task. Specifically, we analyzed how interventions to a model modify the representations of individual points. We applied this analysis in the context of fairness and showed that some debiasing approaches for word embedding do not modify the targeted group of words as expected. *Thus, in Section 5, the application is in choosing the model that impacts the representations the way it is desirable for a specific downstream application.*
>
> Beyond model selection, our aim in identifying unstable points is to emphasize the importance of subjecting these points to further analysis and caution. In Section 4, we showed that there exist points whose representations differ only due to stochasticity present in the models, and that these points are not only more likely to originate from out-of-distribution sources but are also prone to higher misclassification rates. *Thus, in Section 4, the application is in choosing which predictions to trust (i.e. the stable points), rather than choosing which models to use.*
>
> We want to emphasize that such insights are not possible to achieve using global-level similarity measures.
>
> We hope we have addressed all of the reviewer’s concerns and we would happily go into more detail if there are any remaining questions.

---

### Official Review · Reviewer_tyLp · 2023-10-31

**Soundness:** 3 good
**Presentation:** 2 fair
**Contribution:** 2 fair
**Rating:** 5
**Confidence:** 4

**Summary:**

This paper introduces a novel measure for comparing the similarity of latent spaces, with a special focus on its locality property instead of a global one. By independently computing intra-space distances with respect to a specific set of reference points, the authors relate these vectors using the angles between them, assuming a known correspondence between the two sets. The paper then proceeds with three major applications of the method and their validation: i) correlating the measure's similarity output with classification errors on ResNets, ii) out-of-distribution detection, and iii) studying the geometric effects of debiasing techniques on GloVe embeddings.

**Strengths:**

- The paper presents an interesting approach to comparing the similarity of latent spaces and well-motivates its importance;
- The applications proposed, particularly relating to classification errors and out-of-distribution detection on ResNets and the effects of debiasing on GloVe embeddings, are noteworthy;
- The experimental setup appears robust, providing valuable insights that can be considered useful contributions to the field;
- The supplementary material is extensive and also contains the code, enhancing reproducibility;

**Weaknesses:**

- The method explanation lacks some clarity. Multiple readings were necessary due to references to the use of yet-to-be-introduced concepts such as "neighborhood" and "reference points". I think there's also some mismatch between the method presentation and the general take that the neighborhoods of each point are important to their representation since the reference points are never restricted or searched in the neighbors.
- Despite the valuable insights from the experiments, their current setting might not be general enough, limiting broader application and significance. For example, the relationship between PNKA and model disagreement on specific data points is limited to only a cross-training setting, without considering other possible variations such as architectural ones. This is something I would expect in a more theoretical work. The reported results are convincing, but, as commendably acknowledged by the authors themselves in the discussion section, the variation in architecture/tasks is not enough to validate the robustness of the proposed claims;
- There is a noticeable overlap in methodology with cited prior work, particularly Moschella et al. Although the paper frames its method as a kernel method application, it bears a strong resemblance to the direct application of cosine similarity between relative encodings, a technique already explored in the mentioned work. This overlap reduces the novelty of the method but doesn't impact its interesting applications;

**Questions:**

- as described in the weaknesses section, I would ask the authors to please clarify the "neighborhood" concept and its relationship with the reference points;
- In the current manuscript form, I'm recommending a weak reject. However, I'm willing to increase the score if either the lack of variety in experiments or the relationship with previous work is addressed since the former would improve the experimental contributions while the latter the theoretical ones;

---

> ### Author Response · Authors · 2023-11-17
>
> We thank the reviewer for the constructive feedback and comments. Our review responses are placed below.
>
> ## Clarity on PNKA explanation
> We revised the exposition of PNKA to address the clarity issues in the updated version of the paper, in **Section 3**, where PNKA is defined. In summary, **reference points** are used in the computation of PNKA to estimate a point's relative position with respect to other points, within each representation. **Overlap of the neighbors** is another measure by itself, used solely as an intuition behind PNKA's scores. To resolve the ambiguity in the text, we start defining PNKA without referring to these concepts. In the paragraph "Formally defining PNKA", we introduce the idea of comparing the *relative* position of a point with (all) the other points across representations to communicate the formulation of PNKA. Then, in the paragraph "Computing PNKA with stable reference points", we introduce the idea of using a subset of reference points to compute a point's relative position, and we discuss the desirable properties of reference points when analyzing the instability of points. Finally, in the paragraph "Properties", we introduce the intuition behind PNKA by showing the correspondence between PNKA scores and the overlap of neighbors across representations. By making these changes in the text we hope to have made these concepts clearer.
>
> ## Expanding evaluation to other settings
> Our goal in Section 4 is to identify unstable points across two models. We decided to *focus on models that differ by initialization since the representations of these points change by just a stochastic factor*. We agree with the reviewer that this experiment can be extended to other training choices. In fact, in our submission, **Appendix F** includes results on models that differ by architecture, as suggested by the reviewer. In all cases, *we make a similar observation*: points with lower representation similarity are more prone to be misclassified and are more likely to be out-of-distribution (OOD).
>
> ## Addressing differences in methodology from [1]
> We make an initial attempt to clarify this distinction in **Section 1.1.** of the updated version of the paper, and discuss it here as well. Both the method proposed in [1] and ours build on top of the observation made by previous representation similarity measure (RSM) work, e.g. CKA, that models that differ only due to stochasticity tend to have similar representations, and that they position the representations of the same points similarly. However, [1] and our measure differ in how they leverage these observations:
>  1. In our paper, we **focus on those inputs for which the assumption made in [1] does *not* hold** and argue why they are important: [1] makes an assumption, described in Section 3 of [1], that the angles between elements of the latent space are kept the same for *all* elements, which is exactly what we demonstrate in our first result (section 4) as not being the case. Throughout the paper, we expose the importance of studying low-similarity points (i.e., unstable points whose relative position changes).
>  2. The **goal of the papers is substantially different**: In [1], the authors propose using relative representations as a *method* for accomplishing zero-shot model stitching. In our work, we show the importance of studying representations at a pointwise level to obtain a deeper and more fine-grained understanding of DNNs representations. PNKA is just one instantiation of a pointwise representation similarity *measure* that is related to the global CKA measure, however, other measures can also be applied.
>  3. Finally, as also pointed out by the reviewer, **the contributions of both papers are different**: all the experimental setups and findings differ. In [1], the experiments aim to show that relative representations work for zero-shot model stitching. In our work, the experiments aim to show *how important it is to look at representations at the level of individual points*, and we highlight different and important use cases of PNKA. For instance, in Section 5, we specifically focus on comparing how representations of individual points are altered by debiasing approaches – this experiment is meaningless in the context of model stitching.
>
> We hope we have suitably addressed all of the reviewer’s concerns and we would happily go into more details if there are any remaining questions.
>
> References:
>
> [1] Moschella, Luca, Valentino Maiorca, Marco Fumero, Antonio Norelli, Francesco Locatello, and Emanuele Rodolà. “Relative Representations Enable Zero-Shot Latent Space Communication.” ICLR, 2023

---

> > ### Comment · Reviewer_tyLp · 2023-11-22
> >
> > Thank you for your comprehensive response to my initial review. I appreciate the additional clarifications provided and the improvements, especially concerning PNKA in Section 3. I want to apologize for the delayed response on my part. I was also waiting for the perspectives of other reviewers. However, they have not yet provided their feedback.
> >
> > Despite the rebuttal, to me, the distinction in contributions from the methodology presented by Moschella et al. remains unclear. Applying the proposed method to individual data points is a valuable exploration, but in my view, it does not represent a methodological advancement beyond the established framework. Indeed, section 4.2 of Moschella et al. is fully dedicated to using relative similarity as a performance metric. While there's no focus on per-sample properties, the formulation is identical to PNKA without the final average over all the samples.
> >
> > In any case, I want to reiterate that the detailed applications are indeed valuable to the field, so I think the main message of the paper should revolve more around them than on the methodological part.
> >
> > With this in mind, I am keeping my original score but I'm committed to engaging in further dialogue with the Area Chair and the other reviewers in the next phase of the review process.

---

> > > ### Author Response · Authors · 2023-11-23
> > >
> > > We thank the reviewer for engaging with our rebuttal.
> > >
> > > We agree with the reviewer that the most novel contributions of our work lie in the detailed applications of our method. This is also reflected in the fact that a majority of the paper -- (sections 2, 4, and 5) -- focuses exclusively on the applications and *not* the method. We also explicitly and extensively cite related work and acknowledge that we build upon those works. In the paper, we do not make any (strong) claims about the novelty of the method, but at numerous places we emphasize the significance and interestingness of applying the method to better understand representations at the level of individual inputs. So we do feel that the paper is centered around the detailed applications rather than the methodological part (just as the reviewer feels it should). Additionally, we did cite the Moschella et. al. paper in our related work and explicitly acknowledged that our measure is similar to their method. However, we do claim that our goal, contributions, and assumptions are different. So we are unsure where the writing suggests that the primary contribution of the work is the novelty of our methodology.
> > >
> > > We would be happy to change the writing to clarify this point, if the reviewer would point to specific paragraphs or sentences.

---

### Author Response · Authors · 2023-11-17

We thank the reviewers for taking the time and providing useful feedback. We addressed all the questions and concerns raised individually. Here we describe the main revisions and additional experiments incorporated into the revised paper.

## Section 3: clearer explanation of PNKA
Comment raised by reviewer *tyLp* on the difference between the neighborhood concept and reference points. We revised the exposition of PNKA to address the clarity issues in the updated version of the paper, in **Section 3**, where PNKA is defined. In summary, **reference points** are used in the computation of PNKA to estimate a point's relative position with respect to other points, within each representation. **Overlap of the neighbors** is another measure by itself, used solely as an intuition behind PNKA's scores. To resolve the ambiguity in the text, we start defining PNKA without referring to these concepts. In the paragraph "Formally defining PNKA", we introduce the idea of comparing the *relative* position of a point with (all) the other points across representations to communicate the formulation of PNKA. Then, in the paragraph "Computing PNKA with stable reference points", we introduce the idea of using a subset of reference points to compute a point's relative position, and we discuss the desirable properties of reference points when analyzing the instability of points. Finally, in the paragraph "Properties", we introduce the intuition behind PNKA by showing the correspondence between PNKA scores and the overlap of neighbors across representations. By making these changes in the text we hope to have made these concepts clearer.

## Appendix F: varying experimental setup
Comment raised by reviewer *tyLp* on varying the experimental setup, e.g., different architectures. Our goal in Section 4 was to identify unstable points across two models, and we *focused on models that differ by initialization since the representations of these points change by just a stochastic factor*. We agree with the reviewer that this experiment can be done with different training choices. **Appendix F** includes results on models that differ by architecture, and in all cases, we make similar observations.

## Appendix D.1.1: Jaccard distance
Comment raised by reviewer *UEwp* on the relation of Jaccard distance and PNKA. Our primary goal in this work is to highlight the importance of analyzing representation similarity at the granularity of individual data points. To achieve this goal, we designed one instantiation of a pointwise measure, PNKA. However, we note that other pointwise measures can also be employed. We empirically showed in the paper that PNKA is correlated with the overlap of neighbors (**Appendix C.4**), and the Jaccard similarity coefficient (**Appendix C.5**). However, we note that both results are highly dependent on the choice of $k$ being considered. Given the observed variability in the results due to the choice of $k$, we ran an alternative analysis of unstable points using the Jaccard coefficient and appended the results in **Appendix D.1.1** of the revised paper. We see that *the relationship between unstable points and the misclassification rate is highly affected by the choice of $k$ in the Jaccard coefficient*, and the *decision of which $k$ to choose from is not trivial*. Moreover, the optimal $k$ for one architecture and dataset does not generalize to other architectures and datasets. We also show in **Appendix C.2.1** that the number and choice of reference points do not heavily influence the results for PNKA. Thus, Jaccard coefficient cannot provide the same insights as PNKA.

## Use case of PNKA
Comment raised by reviewer *UEwp* on how to use PNKA scores. In Section 5, we show how a model designer can compare different model representations to check which one may be more desirable for a specific downstream task. Specifically, we analyzed how interventions to a model modify the representations of individual points, and showed that some debiasing approaches for word embedding do not modify the targeted group of words as expected. *Thus, the application is in choosing the model that impacts the representations the way it is desirable for a specific downstream application.* Beyond choosing between models, our aim in identifying unstable points is to emphasize the importance of subjecting these points to further analysis and caution. In Section 4, we showed that there exist points whose representations differ only due to stochasticity present in the models, and *the application is in choosing which predictions to trust (i.e. the stable points), rather than choosing which models to use.*

We hope we have addressed all of the reviewers' concerns and we would happily go into more detail if there are any remaining questions.

---

### Meta-Review · Area_Chair_X5tV · 2023-12-01

**Metareview:**

This paper investigates variations in representations at the level of individual data points as the underlying model changes. It empirically shows that models that differ only in initialization result in very different representations for a small set of testing points, which authors deem unstable, and those unstable data instances are also observed to yield a higher likelihood of prediction disagreements. Out-of-distribution points were also often unstable. Authors further investigated adversarially trained models and found those to yield higher representation similarity for a broader range of out-of-distribution points.

While interesting, I agree with some of the reviewers in that the evaluation is a bit limited given that the paper is mostly empirical. The motivation and practicality of the proposal are also questionable. The proposal wouldn't be efficient as an OOD detector, and *unstableness* doesn't seem to be a data property the proposal is able to identify since it's also model dependent. Moreover, adversarially trained models are likely smoother due to how they are trained. It's then expected they'll yield more "stable" representations. All-in-all, the paper needs expanding it's evaluation and better defining its motivation and scope prior to publication.

**Justification For Why Not Higher Score:**

While the paper is quite interesting, there are multiple issues that should be addressed before it being ready for publication as discussed in the latter portion of the meta-review.

**Justification For Why Not Lower Score:**

N/A

---

### Decision · Program_Chairs · 2024-01-16

Reject